# Current Applications and Future Development of Magnetic Resonance Fingerprinting in Diagnosis, Characterization, and Response Monitoring in Cancer

**DOI:** 10.3390/cancers13194742

**Published:** 2021-09-22

**Authors:** Hao Ding, Carlos Velasco, Huihui Ye, Thomas Lindner, Matthew Grech-Sollars, James O’Callaghan, Crispin Hiley, Manil D. Chouhan, Thoralf Niendorf, Dow-Mu Koh, Claudia Prieto, Sola Adeleke

**Affiliations:** 1Imperial College School of Medicine, Faculty of Medicine, Imperial College London, London SW7 2AZ, UK; hao.ding19@imperial.ac.uk; 2School of Biomedical Engineering and Imaging Sciences, St Thomas’ Hospital, King’s College London, London SE1 7EH, UK; carlos.velasco@kcl.ac.uk (C.V.); claudia.prieto@kcl.ac.uk (C.P.); 3State Key Laboratory of Modern Optical instrumentation, Zhejiang University, Hangzhou 310027, China; yehuihui@zju.edu.cn; 4Department of Diagnostic and Interventional Neuroradiology, University Hospital Hamburg Eppendorf, 20246 Hamburg, Germany; t.lindner@uke.de; 5Department of Medical Physics, Royal Surrey NHS Foundation Trust, Surrey GU2 7XX, UK; m.grech-sollars@imperial.ac.uk; 6Department of Surgery & Cancer, Imperial College London, London SW7 2AZ, UK; 7UCL Centre for Medical Imaging, Division of Medicine, University College London, London W1W 7TS, UK; james.ocallaghan@ucl.ac.uk (J.O.); m.chouhan@ucl.ac.uk (M.D.C.); 8Cancer Research UK, Lung Cancer Centre of Excellence, University College London Cancer Institute, London WC1E 6DD, UK; crispin.hiley@ucl.ac.uk; 9Cancer Evolution and Genome Instability Laboratory, The Francis Crick Institute, London NW1 1AT, UK; 10Berlin Ultrahigh Field Facility (B.U.F.F.), Max Delbrueck, Center for Molecular Medicine in the Helmholtz Association, 13125 Berlin, Germany; thoralf.niendorf@mdc-berlin.de; 11Division of Radiotherapy and Imaging, Institute of Cancer Research, London SM2 5NG, UK; mu.koh@icr.ac.uk; 12Department of Radiology, Royal Marsden Hospital, London SW3 6JJ, UK; 13High Dimensional Neurology Group, Queen’s Square Institute of Neurology, University College London, London WC1N 3BG, UK; 14Department of Oncology, Guy’s & St Thomas’ Hospital, London SE1 9RT, UK; 15School of Cancer & Pharmaceutical Sciences, King’s College London, London WC2R 2LS, UK

**Keywords:** magnetic resonance imaging, multiparametric magnetic resonance imaging, prostatic neoplasms, brain neoplasms, abdominal neoplasms, radiotherapy, image-guided, deep learning

## Abstract

**Simple Summary:**

Magnetic resonance fingerprinting (MRF) is a framework for acquiring co-registered multiparametric magnetic resonance mapping with increased scan efficiency. Many studies have explored the use of MRF for cancer management. A review on the current developments in this area has not yet been written but is needed to keep both clinicians and researchers updated. This review summarises recent studies detecting and characterising tumours using MRF, with a focus on brain tumours, prostate cancers, and abdominal/pelvic cancers. Advances in MRF for radiotherapy planning are also mentioned. The principles and limitations of MRF have been simplified to increase accessibility to clinicians with minimal radiological backgrounds. Future oncological applications of MRF are explored, including integrating MRF and deep learning, as well as the use of MRF in assessing disease heterogeneity. We propose further research that needs to take place before MRF can provide a credible means for assessing tumour biomarkers or be accepted by clinicians.

**Abstract:**

Magnetic resonance imaging (MRI) has enabled non-invasive cancer diagnosis, monitoring, and management in common clinical settings. However, inadequate quantitative analyses in MRI continue to limit its full potential and these often have an impact on clinicians’ judgments. Magnetic resonance fingerprinting (MRF) has recently been introduced to acquire multiple quantitative parameters simultaneously in a reasonable timeframe. Initial retrospective studies have demonstrated the feasibility of using MRF for various cancer characterizations. Further trials with larger cohorts are still needed to explore the repeatability and reproducibility of the data acquired by MRF. At the moment, technical difficulties such as undesirable processing time or lack of motion robustness are limiting further implementations of MRF in clinical oncology. This review summarises the latest findings and technology developments for the use of MRF in cancer management and suggests possible future implications of MRF in characterizing tumour heterogeneity and response assessment.

## 1. Introduction

Magnetic resonance imaging (MRI) is a rapidly developing imaging modality with an established and expanding role in the detection and characterisation of many malignancies. A major strength of MRI is the image contrast generated in soft tissues that can be used to provide structural information on patients’ anatomy in a non-invasive way without the risk of ionizing radiation. Specific MRI sequences can detect not only macrostructural information but also characterize tumour cellularity, microstructure and tissue oxygenation [1]

In recent years, there has been a focus on the development of quantitative MRI to improve the objectivity in diagnosis. Different quantitative imaging biomarkers (QIBs) reflect multiple pathological manifestations of disease and thus have the potential to provide more comprehensive non-invasive tumour characterisation. For instance, apparent diffusion coefficient (ADC) may enhance molecular subtyping of breast cancer [2] or increases in tumour T1 relaxation time may indicate better response to anti-angiogenic therapy in ovarian cancer [3]. Post-chemotherapy reductions in T1 relaxation times of murine fibrosarcoma and melanoma models reflect reductions in tumour cell numbers [4] while higher renal cell carcinoma T1 relaxation times are associated with higher collagen volume fractions [5]. Furthermore, higher baseline T2 * values observed in less hypoxic prostate tumours [6] and perfusion fraction measurements from intra-voxel incoherent motion (IVIM) MRI both correlate with tissue fibrosis [7]. Acquisition of different QIBs can be accomplished by a combination of different sequence parameters on MRI protocols. Typically, MRI protocols include multiple sequences that have the potential to yield multiple QIBs from a single scanning session. Unfortunately, each additional sequence increases overall scanning time resulting in protocols that are either too long to be clinically feasible or that have the potential to increase patient discomfort, induce unwanted motion and causing misalignment between different QIB maps. Early quantitative MRI methods rely on exponential fitting routines of signal recovery such as standard inversion recovery (IR) measurement for T1 mapping (e.g., Inversion-recovery-spin-echo [8]) or multi echo spin-echo for T2 quantification (e.g., Carr Purcell Meiboom Gill [9,10]) allowing mapping of various tissue properties. Quantification can also be obtained by analysis of the steady state signal (e.g., T1 Modified Look-Locker inversion recovery (MOLLI) [11] or driven equilibrium single pulse observation of T1 and T2 [12]). However, they are relatively time-consuming and single properties are generated separately as mentioned above. To improve this situation, other single-sequence approaches can provide more than one tissue property, such as inversion recovery TrueFISP [13] (for T1 and T2 mapping), or Multi-Pathway Multi-Echo (MPME) imaging (for T1, T2, T2 *, B0 and B1 + 3D mapping) [14]. TrueFISP however is susceptible to B0 inhomogeneity artefacts on high field MRI scanners [15], while MPME has visibly noisy parameter maps in vivo due to propagated noise from successive processing steps. Another example of simultaneous multiparametric quantitative imaging is MR Multitasking [16], a continuous-acquisition framework that provides multiparametric motion resolved images. Scan times of MR multitasking, however, is not clinically feasible when high resolution (≤1.0 mm slice resolution) is required [16]. Due to the lack of large multicentre studies or readily implementable imaging protocols/sequences, none of the methods mentioned above has achieved widespread clinical acceptance yet.

Magnetic resonance fingerprinting (MRF) uses a different framework to acquire multiparametric maps simultaneously with increased scan efficiency [17]. In contrast to acquiring full-resolution images like conventional quantitative methods do, MRF maps are generated from a precalculated database of possible signal evolutions. Several acquisition parameters are varied during a single MRF scan, leading to a unique temporal signal evolution ‘fingerprint’ because T1, T2, and other sequences are sensitive to the acquisition parameters that are changing. The fingerprints generated are then matched to a precalculated database (‘dictionary’), which is based on existing knowledge of how different tissues behave in a magnetic field. Quantitative parametric maps are then generated by fitting measured signals with predicted dictionary signals on a voxel-wise basis (Figure 1). MRF was introduced by Ma et al., in 2013 [17], and since then, several advances have been made towards establishing clinical applications for this technique. The reduced scan time confers a major advantage relative to other quantitative MRI mapping methods, improving patient comfort and reducing motion artefact [17].

Detailed reviews on the technical aspects of MRF, from Mehta et al. [18] and McGivney et al. [19], covered the physics and engineering principles behind advances made in this area so far. They looked at different methods for reducing matching time, sequence optimization, and dictionary size compression. These reviews offered valuable information for researchers involved in the MRF framework improvement but did not focus on the application potential of MRF in hospital settings. A review from Hsieh et al. [20] discussed the future of MRF in clinical adoption. It summarised some published trials in real patients as supporting evidence and raised important questions about issues such as professional acceptance and lack of standardisation in MRF protocols. While this review covered many potential uses of MRF, its implications on cancer detection and characterization were not mentioned in detail and can easily be dismissed by oncologists. The significance of MRF in oncology should be reviewed in detail and serve as an opportunity to raise more awareness. There are already reviews as such, for instance, looking at cardiac MRF current clinical evidence and a number of challenges to clinical adoption [21,22] (Table 1).

This review summarises recent studies detecting and characterising tumours using MRF, with a specific detailed focus on brain tumours, prostate cancers, and abdominal/pelvic cancers. Literature searches were performed on Medline, Embase, and the Cochrane Library electronic databases for articles published in English from inception until 1 July 2021. References of included studies and reviews were also screened for relevant articles. The following search terms were used (including synonyms and related words): ‘magnetic resonance fingerprinting’ and ‘tumour’, ‘tumor’ or ‘cancer’ or ‘oncology’. Advances in MRF for radiotherapy (RT) planning are also mentioned. The basic principles and limitations of MRF have been simplified to increase the accessibility to clinicians with a minimal radiological background. Future oncological applications of MRF are also explored, including the integration of MRF and deep learning and the use of MRF in assessing disease heterogeneity. We also propose further research that needs to take place before MRF can provide a credible means for assessing tumour biomarkers or be accepted by clinicians.

## 2. The Principles of Magnetic Resonance Fingerprinting

MRF is based on the assumption that different tissues generate unique MR signal evolution when appropriate pulse sequences are applied. This makes its acquisition process fundamentally different from conventional MRI methods (Figure 2). In MRF, the acquisition parameters (such as flip angle, echo time, and repetition time) do not stay the same but are pseudo-randomly varied throughout the acquisition to generate a temporally and spatially incoherent signal output [17] (Figure 2A). The temporal evolution of the signal measured in each voxel (Figure 2B) is then compared with a set of pre-calculated signal evolutions (a dictionary, Figure 2C) that predicts the signal behaviour for the particular acquisition sequence. The dictionary covers a wide range of tissue parameters (e.g., T1 relaxometry, T2 relaxometry, T2 *, etc.) combinations. The best match, for every voxel, between the measured image intensity and all the possible fingerprints in the dictionary (Figure 2D) is then used to generate the quantitative maps (Figure 2E). In the case of the MRF framework proposed by Ma et al., Bloch equations were used to calculate the temporal evolution of the signal [17]. However, this is not the only option, and other approaches, such as the extended phase graph (EPG) formalism, can be employed to improve computation efficiency [26].

A characteristic feature of MRF is that the images produced for each time point are usually highly undersampled (Figure 2B). Initial studies suggested that the presence of undersampling artefacts in these temporal images is not critical as long as they are not spatially or temporally correlated, because it is their temporal evolution, and not the individual quality of the images that is used to produce the voxel-wise multiparametric quantification by dictionary matching [17]. Early approaches attempted to use zero-filled reconstruction to produce the images for each time point [17], however subsequent studies demonstrated that remaining aliasing artefacts can affect the multiparametric quantification [27]. Thus, several undersampled reconstruction approaches have been proposed exploiting the temporal and spatial redundancies in the highly undersampled time series of images [28,29,30].

## 3. Magnetic Resonance Fingerprinting for Imaging Cancer

### 3.1. Brain Tumours

Early detection of metastatic and malignant brain tumours on imaging helps support management decisions and can give important prognostic information early on during treatment [31]. Tumour cell characterisation is important for decision making and can potentially impact patient outcomes [32]. Conventional MRI with intravenous gadolinium-based contrast is used as a first-line imaging modality when metastatic lesions are suspected as this increases sensitivity, but MRI has limited specificity for differentiating between primary glioblastomas (GBMs), lower grade gliomas (LGGs), and brain metastases, [33]. Although other imaging techniques, such as perfusion imaging or PET, can help differentiate between brain tumours grades and origins [34,35], quick quantitative imaging modalities without exposure to ionizing radiation or contrast agents are still desirable. Co-registered T1 and T2 mappings from MRF can be beneficial for tumour cell characterisation and treatment management. However, there is still the need for advanced post-processing methods to surmount computational challenges. The T1 value is known to correlate with brain water content [36,37]; it can be used in particular in the assessment of perilesional oedema. Lesional T2 mapping has been associated with the early detection of tumour progression under anti-angiogenic therapy [38] and the potential to differentiate between molecular subtypes of grade II and III gliomas [39].

A preliminary study from Badve et al. suggested that MRF-derived T2 relaxation times from solid lesion components are higher in LGGs than in metastases (Figure 3) [40]. Differences in MRF-derived T1 relaxation times of peritumoral regions of GBMs and LGGs have also been identified [40]. In a later study, using the same data set, radiomic texture analysis was applied to further improve the differentiation accuracy [41]. MRF maps after texture analysis also showed the ability to predict patient survival time in the GBM cohort [41]. Considering the small sample size (31 patients), more evidence is needed to prove the diagnostic and prognostic reliability and repeatability of MRF. Similar studies were also conducted on paediatric and young adult patients [42]. MRF-derived T1 and T2 maps were significantly different for normal-appearing white matter, solid tumour, and peritumoral regions. MRF in children could help avoid potentially harmful, long-term retention of gadolinium-based contrast agents. The use of MRF could further avoid the risk of contrast allergy and minimises the need for sedation in children. The use of MRF has also been investigated in meningiomas. In a recent study, meningothelial meningiomas showed significantly higher T1 and T2 values than transitional and fibrous meningiomas on MRF images [43], whereas conventional MRI imaging and ADC values collected from the same set of patients detected no statistically significant differences between these tumour subtypes. Although larger cohorts are needed to validate the findings in these proof-of-concept studies, MRF is a promising potential QIB for investigating brain tumours. Repeatability and reproducibility of MRF for healthy brains have been tested under 1.5T and 3T at two different centres. Excellent repeatability and good reproducibility were reported [44]. More multicentre studies will be needed to establish the use of MRF more widely in common clinical settings, and parametric maps acquired by different MR vendors should also be compared.

Other than detection, MRI is routinely used to define lesion boundaries of various cancers for RT planning [45]. However, the signal intensity of MRI images is expressed in arbitrary units and qualitative in nature. Conventional MRI also has low repeatability and reproducibility for disease monitoring due to the variability in acquisition parameters [46,47]. There is an increasing interest to use quantitative MR data in RT treatment decision making, toxicity assessment and response monitoring. Using MRF to produce reliable relaxometry maps has been previously described [44]. However, the hardware and configuration of MRI systems in a radiation oncology department are different from those used for diagnostic imaging. Diagnostic MRI scanners use coils designed to maximise image quality, whereas MRI scanners for guiding RT use flexible body coils combined with specific coil frames to allow patient-immobilization devices to be placed and to maximize geometric accuracy. Thus, studies on MRF for cancer diagnosis with diagnostic MRI scanners may not be entirely applicable to MRI scans in RT. Lu et al. have used fast imaging with the steady-state precession (FISP) MRF to obtain T1/T2 maps using a scanner setup for intracranial tumour RT treatment planning [48]. A combination of channel flexible body and head coil was designed, with T1 and T2 mapping data reported to be reliable and repeatable with intra-/inter-scanner intensity variations (The intra- and inter-scanner variability of the intensity-normalized MRF T1 was 1.0% ± 0.7% and 2.3% ± 1.0% respectively). Due to the effect of geometric distortion, correct T1 and T2 values were retrieved using template matching. Despite the promising results from this initial attempt of guiding RT treatment with MRF, the issue of non-real-time reconstruction after data acquisition in MRF is a challenge that needs to be resolved.

MRF can also measure additional parameters beyond T1 and T2 relaxivity. Several studies have also explored the possibility of measuring Chemical Exchange Saturation Transfer (CEST) with MRF [49,50,51], as CEST has shown potential in monitoring treatment response in brain tumours [52,53,54]. MRF has the potential to greatly reduce CEST scan time compared to conventional methods, making CEST more suitable for the clinical setting. Whole brain images using 3D-MRF with high resolution are being developed which can be helpful for volumetric analysis or microvascular analysis in the future [55]. Lemasson et al. looked at microvascular properties of brain tumours in rats using MR vascular fingerprinting. Transition to human application might be possible with improving algorithm design and mathematical models [56]. A recent study using PET-MR imaging with MRF showed promising results for separating low-grade and high-grade gliomas [57]. This modality has also been shown in preliminary data to be useful for detecting molecular changes and genetic mutations in brain tumours. MRF was used in this study in addition to conventional MR sequences to provide fast quantitative T1, T2, static magnetic field (B0) inhomogeneity and proton density (M0) mapping.

### 3.2. Prostate Cancer

MRI is the standard of care imaging modality for diagnosing prostate cancer. Multiparametric MRI is recommended by the National Institute for Health and Care Excellence (NICE) as the first-line investigation method for suspected localised prostate cancer, and the results are reported using a 5-point Likert scale based on radiologists’ overall interpretation of the scan [58]. Separate assessment of individual sequences is not required, and this approach arguably lacks reproducibility, especially with less experienced radiologists, and can undermine overall accuracy [59]. An alternative guideline—the Prostate Imaging Reporting and Data System, version 2 (PI-RADS™ v2)—assesses T2-weighted imaging, diffusion-weighted imaging (DWI), and dynamic contrast-enhanced (DCE) imaging separately and then in combination. However, both guidelines are qualitative, and there is a growing demand for more objective quantitative data. T1 and T2 mapping from MRF can provide a fast solution for more accurate assessment. Baur et. al. used MOLLI sequences to map prostatic lesion T1 values and reported that T1 relaxation time differs significantly between prostate cancer and benign prostate tissue, with lower T1 in cancerous lesions [60]. In a systematic review by Lee, 11 out of the 17 included studies reported T2 relaxation times of normal peripheral zone (range 111.6–169.6 ms) being higher than those of prostate cancer (range 67–109 ms), with benign prostatic hypertrophy likely contributing to the overlap [61].

An initial retrospective study from Yu et al. [62] showed that a 3.0-T unit MRF combined with standard ADC mapping had high sensitivity for differentiation between peripheral zone prostate cancer and normal prostate tissue (Figure 4). The acquisition time of MRF was also shorter than the acquisition time for standard clinical sequences used in that institution when acquiring T1, T2, and ADC maps (average of 7.5 min versus 21 min). T1, T2 signal intensities and ADC values reported were all lower in cancers compared with normal peripheral zones in the prostate, with reported mean cancer T1 of 1628 ms (±344) and mean T2 of 73 ms (±27), in contrast to 2247 ms (±450) and 169 ms (±61), respectively, for a normal gland. The T1 value changes detected with MRF had not been previously reported, possibly due to difficulties in measuring T1 signal intensity changes on conventional weighted MRI images. These findings were also in agreement with the study results from Panda et al. in which biopsy was used as the reference standard [63]. Panda et al. also reported that T2 and ADC values together could separate clinically significant cancer from low-grade cancer in the peripheral zone. Another retrospective study by Panda et al. showed that T1/T2 MRF combined with a separately acquired ADC mapping may improve lesion characterization in the transition zone of the prostate [64]. MRF T1 and ADC values were found to be complementary for malignant lesion staging, whereas there was an overlap in MRF T2 values between different lesion types [64]. Han et al. recently used 3D MRF in phantoms and 90 patients with suspected prostate cancer [65]. Findings were consistent with the results from 2D MRF trials for prostate cancer, and the image acquisition protocol provided faster coverage of the entire prostate gland (4 min rather than 7–8 min). Considering that the studies cited above were from individual institutions, multi-institutional datasets with bigger sample sizes will improve the generalisability and repeatability of the findings. However, the consistent results obtained from separate independent preliminary trials are encouraging. High inter-scanner reproducibility of MRF T1 relaxometry was reported for assessing healthy prostate recently for both 1.5T and 3T MRI scanners [66]. Similar studies in patients with suspected prostate cancer have yet to report their findings. To date, MRF has been used for the characterization of prostate lesions but not detection as a higher resolution is required for structural T2-weighted imaging. This modality may also benefit from the development of simultaneous MRF mapping of T1, T2, and ADC to eliminate the risk of misalignment. Jiang et al. inserted multiple magnetization preparation modules in a FISP-based MRF sequence to achieve simultaneous T1, T2, and ADC mapping in less than 60 s per slice [67]. Prostate imaging may benefit from further developments of this novel MRF sequence. The feasibility of guiding RT or monitoring treatment response for prostate cancer using MRF has also yet to be investigated.

### 3.3. Lesions in Abdomen and Pelvis—Liver and Ovaries

Lesion detection and characterisation within abdominal and pelvic organs have the potential to benefit significantly from a time-efficient, free-breathing, quantitative parametric mapping offered by MRF. Chen et al. used FISP T1/T2 MRF to acquire fast abdominal images (19 s for each section) [68]. Six patients with focal liver lesions were imaged and the longer MRF T1 and T2 relaxation times in metastatic tumours were consistent with previous non-MRF findings [69,70]. Powered studies with different lesion and tumour types are needed for investigating the potential of this method in lesion detection and characterization. Three-dimensional acquisitions of the abdomen are still restricted by imaging speed and respiratory motion. A combination of MRF abdominal imaging with parallel imaging and simultaneous multi-slice acquisitions has the potential to increase coverage within a shorter time frame [55,71]. Huang et al. recently integrated MRF and a pilot tone navigator with retrospective gating to successfully obtain 3D abdominal MRF images in free-breathing healthy volunteers [72].

Feasibility MRF studies have also been conducted in the pelvis. Kaggie et al. investigated simultaneous T1, T2, and relative proton density mapping for ovarian cancer using MRF [73]. Ascites is easy to detect, but in large quantities, it can introduce artefacts, particularly at high field strengths. These artefacts from standing waves are caused when conductive ascitic fluid attenuates the radiofrequency fields, resulting in signal loss [74,75]. In vivo, the presence of ascites did not pose challenges to MRF (Figure 5g–i) if the T2 detection range was wide enough. Figure 5a–c shows biological variation on the MRF images, which might be important for future studies evaluating intratumoral heterogeneity [76]. Further validation studies in larger cohorts are still warranted.

MRF also showed potential for RT motion management and more detailed information on lesion extent indicates a potential role for RT planning. Li et al. proposed the concept of time-resolved MRF with repeated acquisitions using an unbalanced FISP and spiral-in–spiral-out trajectory for dealing with motion [77]. This technology was tested on both phantom and healthy volunteers and was capable of imaging respiratory motion with simultaneous T1 and T2 mapping of the abdominal organs. To the best of our knowledge, no MRF studies evaluating abdominal cancer treatment response have been reported.

## 4. Potential Future Developments for Cancer Management

As discussed above, MRF can be used to acquire multiple parameters simultaneously in a shorter acquisition time. Therefore, progress in MRI for cancer characterization and treatment can potentially be tested on MRF in addition to other techniques/modalities as illustrated in Figure 6 to improve lesion detection and cancer screening. MRF may also be integrated with other new technologies. For instance, the possibility of integrating MRF, super-resolution reconstruction and nanoparticles to diagnose small pancreatic cancer lesions is being explored [78]. Current clinical studies for MRF focused more on the feasibility and reliability of using MRF in tumour detection usually at single centres [40,56,63,64,68,73] (Appendix A) overview of published MRF oncological application studies). It will be interesting to see the comparison between MRF and other established imaging modalities in future studies as well as reproducibility studies between sites, vendors and patient cohorts. However, to do so, limitations regarding storage space needed by dictionaries and post-processing time must be further addressed. For this purpose, the use of artificial intelligence might be a promising approach.

### 4.1. Inter- and Intra-Tumoral Heterogeneity

When using MRI to differentiate malignant from benign tumours, heterogeneity (as described in Figure 7) is an important feature to evaluate, alongside tumour size or volume [85]. Within the same bulk of a tumour, different clones of cancer cells behave differently and acquire a unique set of genetic mutations which perpetuates their survival [86]. These Darwinian evolutionary changes are often due to adaptive or replicative pressure and have been linked to the development of treatment failure and resistance [87]. A promising technique that could characterise changes or differences between metastatic sites or within a single site without the need for invasive biopsies will be welcome [88,89]. Ability to better characterise tumoral heterogeneity could enable more effective treatment planning [90]. There has been particular interest in using DWI and DCE MRI which measure water diffusion, vascular fraction (v_p_) or tumour permeability. Different values acquired in different regions of the tumour may indirectly reflect variations in the expression of tissue growth factors and anti-angiogenic factors hence indicate heterogeneity [91,92]. Since the output of MRF is multiparametric, T1 and T2 measurements can potentially be produced in combination with ADC mapping or v_p_ value to characterise lesion heterogeneity. Currently, biopsy remains the gold standard for many types of cancer diagnosis and characterization [89,93]. As an invasive procedure, tissue biopsy has a clear risk of complications [88]. Studies have suggested multi-parametric MRI might allow up to 27% of patients to avoid a biopsy for possible prostate cancer though the specificity of MRI could still further improve [94]. MRF has the potential of being a validated and repeatable imaging tool for lesion characterisation and further reduces the burden of biopsy. MRI is also used in monitoring treatment responses and potential applications of MRF in this aspect should be explored.

### 4.2. Response Monitoring

Response valuation criteria in solid tumours (RECIST) 1.1 remains the reference standard for assessing disease response in many malignancies but is fraught with limitations. For instance, it has been categorically shown not to be useful in bone lesions, except when there is a measurable extra-osseous soft tissue component [95]. It is also unable to assess diseases such as inflammatory breast disease, leptomeningeal disease, cystic lesions, and even nodal or soft tissue lesions smaller than 1 cm [95]. Ability to absolutely quantify changes in signal characteristics to detect response much earlier than size change will be of major clinical significance. DWI has been investigated as a biomarker for detecting tumour response. Compared with standard response criteria, such as the RECIST, it is sensitive to changes at the cellular level prior to changes in gross tumour size. However, lack of standardisation and susceptibility to artefacts and image distortion remains problematic. When analysed in combination with other parameters measured by MRF, a more accurate evaluation of tumour progression might be achieved [92,96]. With more quantitative information being collected by MRF, personalised medicine and treatment plans can potentially be developed for individual cancer patients [97]. Study by Bruijnen et al. also explored the feasibility of gradient spoiled 2D T1/T2 MRF on hybrid MRI-linac systems to assess tumour response to radiotherapy [98]. However, work in this area is still in the early stages. MRI-linac systems are now commercially available, and active research is ongoing for their increased integration into the radiotherapy treatment workflow [99]. Image-guided radiotherapy is now increasingly used to identify organs at risk and to keep the radiation dose to them as low as possible [100]. Despite MR-linac availability for clinical use, many centres still rely on pre-treatment acquired computerised tomography (CT) or synthetic CT images to get Hounsfield values. The online MRI acquired then needs to be registered to the treatment planning data and becomes a problem when there is random motion during treatment as real-time treatment planning is required. MRF has the advantage of promising full quantitative output, which could potentially be used instead of Hounsfield values acquired from CT. On-boarding of MRF into MR-guided or MR-linac systems could radically change the planning of radiotherapy treatment.

## 5. Current Limitations of MR Fingerprinting

Although the emerging clinical evidence has demonstrated that MRF can provide more rapid and specific tissue properties for tumour characterization, there are still many limitations hindering wider implications of this framework. Susceptibility to motion artefact can impair the accuracy of mapping and image reconstruction times need to be shortened significantly for the potential use of MRF in hospital settings.

### 5.1. Motion Robustness

Subject movement during acquisition is one of the main challenges for accurate MRI quantitative mapping. Ma et al. demonstrated in their original work [17] that the MRF framework can tolerate non-periodic abrupt motion towards the end of the scan; however, more subtle or natural motion patterns, such as breathing or cardiac movement, could hinder MRF quantification, especially in the case of through-plane motion [101]. Yu et al. [102] showed that T2 value can be systematically underestimated due to through-plane motion in the middle of the scan when a large flip-angle is applied. Some sources of motion, such as breathing, may be partially avoided by performing short MRF scans during a breath-hold; however, bulk motion can be difficult to avoid in patients with certain neurological diseases. Such challenges are greater by several orders of magnitude in the abdomen and pelvis. In order to achieve motion compensation or correction in 2D MRF, many approaches have already been proposed [103,104,105,106]. Mehta et al. developed motion-insensitive MRF reconstruction algorithm that increases sensitivity of MRF to rigid-body motion through iteratively performing dictionary matching, motion estimation and correction, and image reconstruction [103]. Cruz et al. used sliding window reconstruction followed by image registration to estimate motion and correct acquired k-space data. Two-dimensional images reconstructed with low-rank inversion had fewer in-plane motion errors [101]. This method was further validated by Xu et al. [104] with both simulated and in vivo experimental evidence. Three-dimensional MRF, which has a potential for monitoring brain tumours, is also sensitive to motion artefacts, and effort has been made to improve motion robustness of MRF in this direction, too. Cao et al. have shown that 3D MRF, multi-axis spiral projection imaging (maSPI) is more motion-robust compared with stack-of-spiral (SOS) imaging [107]. Based on this finding, Kurzawski et al. successfully used random trajectory ordering and separate image reconstruction for each 7 s timescale segment to obtain less coherent artefacts [108]. Cruz et al. employed respiratory bellows driven localized autofocus for beat-to-beat translation motion correction when obtaining 3D mapping of the whole heart in a single free-breathing scan [32]. Recently, Huang et al. integrated MRF with pilot tone navigator to produce 3D abdominal imaging without the need for breath holding [72]. More studies still need to be carried out to improve the motion robustness in 2D and 3D MRF, especially for through-plane motion in 2D MRF.

### 5.2. Acquisition and Processing Time

One of the advantages of MRF is that the number of parameters quantified from a single scan can be extended by cleverly designing the acquisition sequence and accurately predicting signal evolution. Original MRF work showed simultaneous quantification of T1, T2, M0, and B0, but since then, many other parameters of interest for oncological and non-oncological applications have been mapped, such as MRF for simultaneous T1, T2, and T2 *; fat fraction mapping for liver; and T1, T2, and diffusion parameter mapping for brain, prostate, heart, ovary, and abdomen [109]. Mapping additional parameters in MRF, however, usually requires adding extra dimensions to the dictionary, which increases dictionary size exponentially and poses computational challenges, including the need for increased storage space, especially for cardiac or vascular MRF, where patient-specific dictionaries must be created taking into account the heart rate measured during the scan. As an example, in a 4 min abdominal MRF scan by Serrao et al., 4 gigabytes of disk space were used because of the near continuous data acquisition and the large amount of coil channels [110]. Dictionary extension also directly increases the post-processing time needed for fingerprint matching and dictionary development [19]. Therefore, for potential clinical use, post-processing time should be further reduced to avoid a delay between patient positioning and imaging review by scanning radiographers and radiologists. To address this, many ideas have been proposed. One of them is the use of a compressed dictionary via singular value decomposition (SVD), where the dictionary dimension is reduced to a low-rank subspace, and most of the information is still retained from the original dictionary—fewer points are then needed, and hence, the dictionary matching time is reduced [28]. In the last years, SVD and dictionary compression have helped to reduce not only matching times and dictionary size, but also reconstruction time and undersampling artefacts [111,112,113,114].

The emerging transition from 2D MRF to 3D MRF with higher spatial resolution and higher signal-to-noise ratio also faces the problem of long acquisition and processing times [19]. Approaches have been developed to shorten acquisition time; for example, Ma et al. undersampled individual time points and shortened the waiting time between groups to acquire 3D T1, T2, and M0 whole brain maps in less than 5 min [115]. Improved accuracy with a spatial resolution of 1.2 × 1.2 × 2 mm^3^ was achieved by Liao et al. with a whole-brain scan (~19 cm volume) in ~3.5 min [116], when they accelerated the stack of spirals 3D acquisition by using sliding window reconstruction with generalized autocalibrating partially parallel acquisitions (GRAPPA). However, the reported 20-h reconstruction time still makes the MRF prohibitive to use in common clinical settings.

Development of neural networks and deep learning algorithms in recent years has presented the opportunity to reduce post-processing time and storage space needed for MRF dictionaries. Chen et al. [55] used parallel imaging with deep learning techniques for producing whole-brain 3D MRF images in a ~7 min scan with a spatial resolution of 1 mm^3^. The processing time was shortened seven-fold using deep learning as opposed to standard template matching [55]. Alternatively, work from Oksuz et al. [117] showed simple recurrent neural networks (RNNs) can predict the T1 and T2 values with high accuracy in less than 20 ms once RNNs are trained for an hour. The MRF deep reconstruction network (DRONE) developed by Cohen et al. improved the reconstruction process considerably [118]. The trained neural network reconstruction function was 20 times smaller than conventional MRF dictionaries and reconstruction time was ~300–5000-fold shorter than with conventional methods, while training the network with a dictionary of ~69,000 entries took approximately 10 to 74 min on an Nvidia K80 GPU with 2 GB of memory.

### 5.3. Adoption of MR Fingerprinting by the Imaging and Oncology Community

Large scale multi-centred trials across different vendors are needed to provide the imaging and oncology communities with enough evidence that MRF can be used for accurate, repeatable, and reproducible diagnosis and tumour characterisation. According to the imaging biomarker roadmap for cancer studies prepared by O’Connor et al., parameters measured by MRF still need to cross the first translational gap with more technical and clinical validation [119]. The evidence for reliable MRF measurements is not yet sufficient to establish its role as a ‘medical research tool’. For MRF data to be used in ‘clinical decision-making tools’, their cancer screening, diagnostic, and predictive abilities need to be tested on patients in different centres. Radiologists, diagnostic, and therapeutic radiographers will need to be familiar with the new image registration methods and scanning protocols. MRF packages used in all the studies referenced in this review were provided by the vendors because investigating the oncological use of MRF is still at a preliminary stage. However, a vendor-neutral MATLAB-based software MRF package has recently been published on GitHub [120], and, hopefully, similar resources will be accessible for all researchers and clinicians in the near future. Reconstruction of MRF can be carried out on a standard vendor’s computational hardware and some data post-processing can be performed on a laptop or desktop. However, producing high-resolution mapping within a clinically relevant timeframe is still challenging. Large-scale implementation of MRF requires adequate post-processing pipelines to be set up in hospitals. Cost and health economic implications of MRF compared with standard methods will also be important to adoption either as a replacement or complementary approach alongside current standards [20].

## 6. Conclusions

We described advances made by conventional MRI and how quantitative MRI parameters are potential biomarkers for many cancer types. We have introduced the concept of magnetic resonance fingerprinting and how it has been able to lead to time-efficient quantification of imaging features seen on MRI, in absolute terms. This has implications beyond just anatomical or morphological features seen on imaging, but it could advance radiotherapy treatment planning, tumour response monitoring, and evaluation of disease heterogeneity. Future use of MRF could also span into imaging genomics, which involves mapping and matching biological and imaging features with the genomic landscape and other tumour characteristics. Furthermore, if the MRF-derived dictionaries are standardised across scanners/institutions, this will lead to exciting opportunities to carry out cross-centre or cross-vendor research beyond what is currently achievable. A multi-speciality collaboration is required in order to make MRF ready for clinical primetime in cancer in the near future.

## Figures and Tables

**Figure 1 cancers-13-04742-f001:**
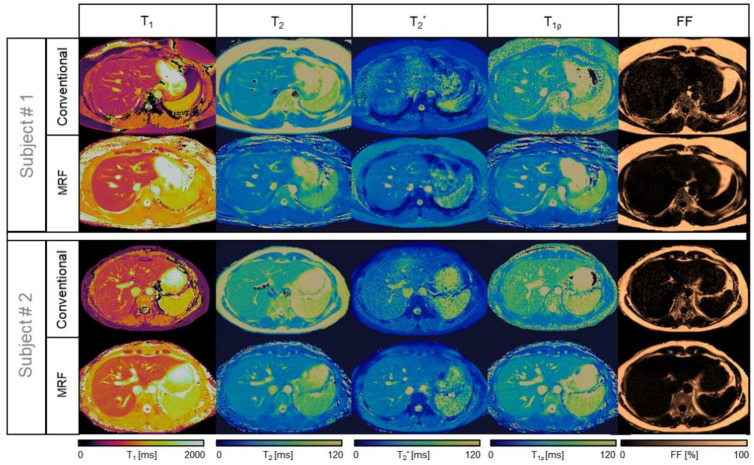
Comparison between liver MRF maps and conventional images from two healthy subjects. The conventional referencing sequences are (**top** row, left to right for each subject) T1-MOLLI, T2-GraSE, 8-echo GRE T2 *, T1⍴-TFE, and 6-echo GRE FF, compared against the proposed inherently co-registered T1, T2, T2 *, T1ρ, and FF abdominal MRF maps (**bottom** row left to right for each subject) obtained from a single 18 s scan.

**Figure 2 cancers-13-04742-f002:**
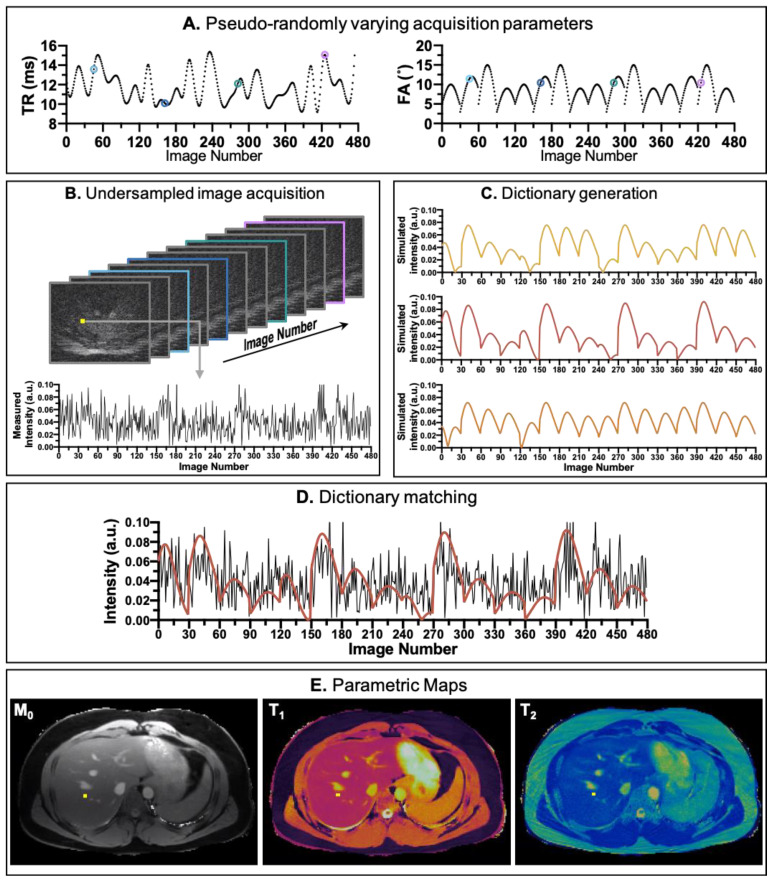
An overview of the MRF framework. (**A**) randomised series of repetition time (TR) and variable flip angles (FA) used for acquisition. (**B**) example of undersampled images acquired (**C**) example of dictionary entries (**D**) matching the temporal evolution of the signal measured with the dictionary (**E**) Abdominal MRF maps generated from the matching process with the dictionary.

**Figure 3 cancers-13-04742-f003:**
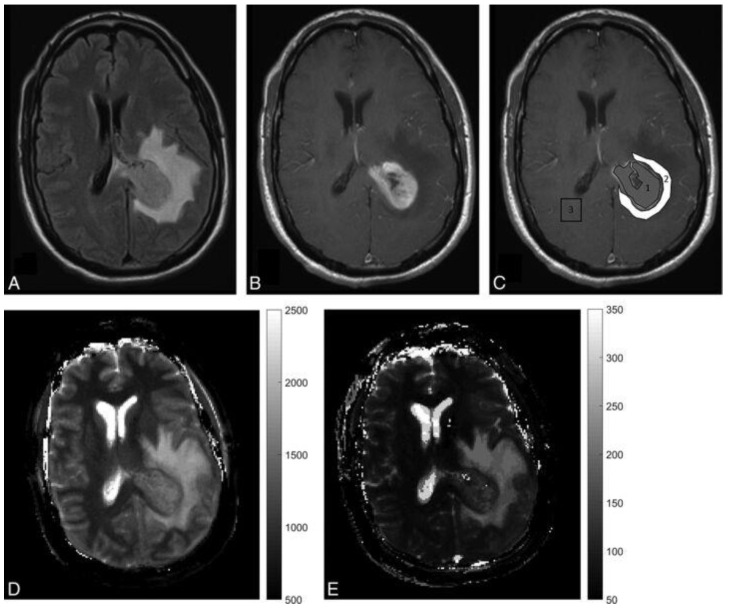
A 45-year-old male presenting with severe headaches and altered sensorium was later diagnosed with glioblastoma. (**A**,**B**) are FLAIR- and contrast-enhanced T1-weighted images from the clinical scan, which demonstrate a left parietal enhancing lesion with peritumoral FLAIR hyperintensity. (**C**) is a post-contrast T1-weighted image with ROI overlay. 1 shows a solid enhancing tumour region, while 2 shows a peritumoral white matter region. 3 in contralateral hemisphere denotes the contralateral white matter measurement. (**D**,**E**) are MRF-derived quantitative T1 and T2 maps showing a wider extent of disease. Reprinted with permission from ref. [40]. Copyright 2020 American Journal of Neuroradiology.

**Figure 4 cancers-13-04742-f004:**
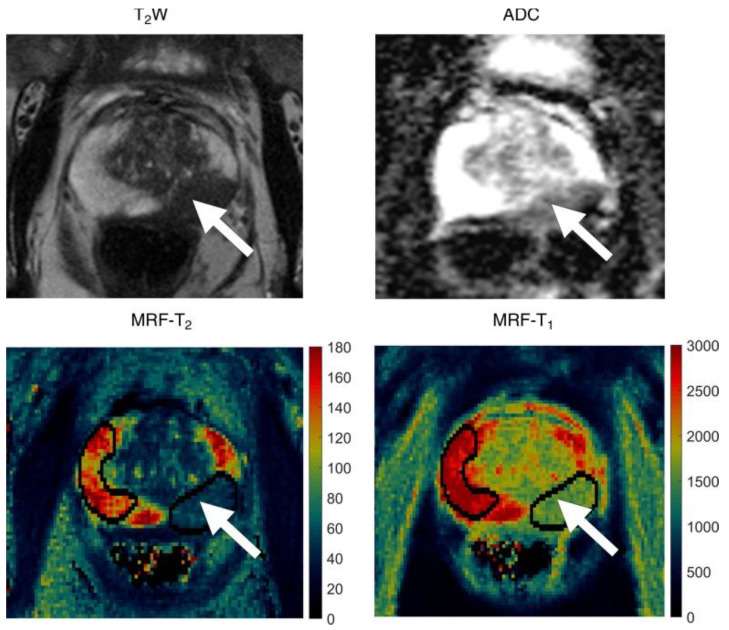
Images in 72-year-old man referred for an elevated prostate-specific antigen of 9.87 ng/mL with minimal urinary symptoms. Patient underwent limited MR imaging and targeted biopsy of lesion in left mid prostate. Prostate adenocarcinoma with Gleason score 4 + 3 = 7 was diagnosed at cognitively targeted biopsy. T2-weighted image (T2W), ADC apparent diffusion coefficient map, MR fingerprinting (MRF)—T2 map, and MR fingerprinting—T1 map show corresponding hypointense lesion in left mid prostate (arrow) and NPZ normal-appearing peripheral zone in right hemi-prostate. Reprinted with permission from ref. [62]. Copyright 2020 The Radiological Society of North America (RSNA).

**Figure 5 cancers-13-04742-f005:**
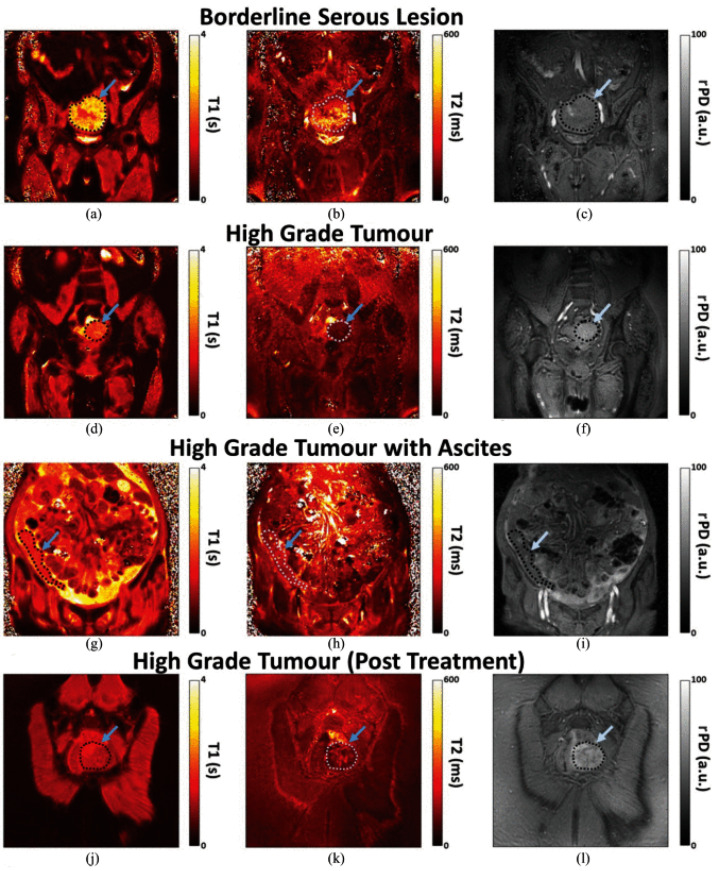
Coronal MRF T1, T2, and PD quantitative maps of the four patients with (**a**–**c**) borderline serous and (**d**–**l**) high grade tumours. Images (**a**,**b**) show areas of variation in tumoral signal intensities and hence possible disease heterogeneity. HGSOC tumours with ascites was visible on MRF images, arrowed regions on (**g**–**i**). One of the patients who had treatment response, MRF (**j**–**l**), was later confirmed to have no histological evidence of residual disease [76].

**Figure 6 cancers-13-04742-f006:**
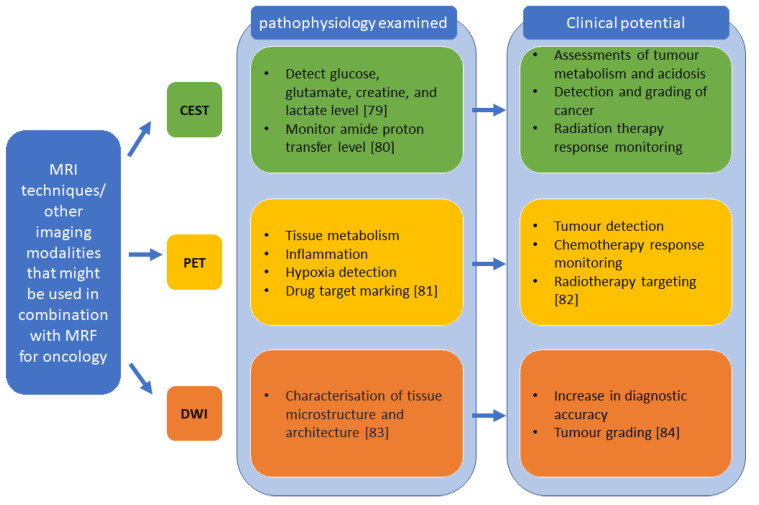
MRI techniques and other imaging modalities that could be explored in combination with MRF in oncology [79,80,81,82,83,84].

**Figure 7 cancers-13-04742-f007:**
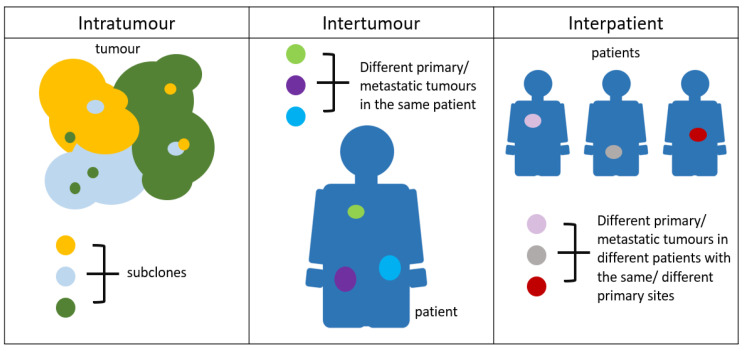
Within the same tumour, different subclones often exist. These clones may demonstrate distinct biological properties and hence disease aggressiveness. Even carefully planned biopsies may fail to target, acquire, and characterize these different cellular or tissue subtypes. Novel imaging techniques are better able to surmount this challenge, non-invasively. Different sites of disease within the same patient (intertumoral) and patients with the same histologically tumour type may have very distinct tumour biological characteristics (interpatient), and hence, this could impact their outcomes. Novel, next-generation imaging offers a holistic non-invasive anatomical, functional, and biological evaluation of disease.

**Table 1 cancers-13-04742-t001:** Existing reviews on MRF with their areas of focus compared.

Title	First Author	Journal	Year of Publication	Focused on: Technical/ Clinical/Both	Focused on: Cancer/Non-Cancer/Both
Cardiac magnetic resonance fingerprinting: technical developments and initial clinical validation [21]	Gastao Cruz	*Current Cardiology Reports*	2019	both	non-cancer
Cardiac magnetic resonance fingerprinting: technical overview and initial results [22]	Yuchi Liu	*JACC: Cardiovascular Imaging*	2018	technical	non-cancer
Cardiac magnetic resonance fingerprinting: trends in technical development and potential clinical applications [23]	Brendan L Eck	*Progress in Nuclear Magnetic Resonance Spectroscopy*	2021	both	non-cancer
Magnetic resonance fingerprinting Part 1: Potential uses, current challenges, and recommendations [24]	Megan E Poorman	*Journal of Magnetic Resonance Imaging*	2019	both	both
Magnetic resonance fingerprinting review Part 2: Technique and directions [19]	Debra F McGivney	*Journal of Magnetic Resonance Imaging*	2019	technical	non-cancer
Magnetic resonance fingerprinting: a technical review [18]	Bhairav B Mehta	*Magnetic Resonance in Medicine*	2018	technical	non-cancer
Magnetic resonance fingerprinting: an overview [25]	Ananya Panda	*Current Opinion* *in Biomedical* *Engineering*	2017	both	both
Magnetic resonance fingerprinting: from evolution to clinical applications [20]	Jean J L Hsieh	*Journal of Medical* *Radiation Sciences*	2020	both	non-cancer

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
