# Peer review of "Current Applications and Future Development of Magnetic Resonance Fingerprinting in Diagnosis, Characterization, and Response Monitoring in Cancer"

_cancers, 2021, doi:10.3390/cancers13194742_

Round 1
Reviewer 1 Report
The paper reviews the use of MRF in cancer studies. The manuscript is well organized and most of the literature papers on the subject are referenced without biases. I have only minor comments:
-It seems to me that section 4.1 on deep learning should appear in section 5.2 as a possible solution for correcting long acquisition and processing times, rather than a potential future development for cancer management.
-In a few places in the manuscript (lines 284-286 and lines 328-329), it would be interesting to provide numerical values regarding MRF scan times.
-On the same topic, I wonder if the comparison between standard MRI and MRF scan times (23min vs 14s) in lines 106-110 is fair in terms of number of slices acquired as well as spatial resolution.
-I had troubles understanding the following sentences:
Line 100-101: “lack clinical feasibility at where high resolution”
Line 346: “shows possible biological variation is visible”
Line 381-382: “and space needed by using machine learning”
Line 456-457: “there are still many limitations need to be”
-There seems to be a problem with the references around ref 56-59 where the names of the authors in the main text don’t correspond to the ones in the reference section.
-Figure 6 is never called or described in the text.
-Typo line 349: MRF also “showed”
Reviewer 2 Report
The authors introduce MR Fingerprinting (MRF) as a relatively new imaging framework capable of providing accurate and simultaneous quantification of multiple tissue properties for improved tissue characterization and disease diagnosis. The narrative review focuses on mostly clinical applications in oncological disorders and is supposed to be addressed at physicians with minimal radiological background knowledge. The authors introduce a hybrid format combining a review of reviews with a review of literature.
Strenghts:
A large body of literature is included in this full review. Potential future developments and current limitations are discussed.
Weaknesses:
The review appears very convoluted which makes it hard to read. As the authors state, there are many high quality and well-structured recent MRF reviews, some of them focused on oncological applications as well. The introduction should put a stronger emphasis on the new angle and discuss possible limitations of past reviews. The review should then focus on the new angle and possible limitations.
The review fails to mention which keywords and database sources were used searching the literature. Up to which point in time have the authors been scanning the literature?
A review aimed at physicians with minimal radiological background knowledge or even at mostly clinically oriented radiologists should explain the technical principals of MRF in more basic terms from signal acquisition and which parameters can be simultaneously mapped, to dictionary generation (describe it simpler such as “pre-calculated database”; The reader will want to know if it is patient specific!), and finally to the process of pattern matching within the dictionary and assignment of the properties of all fingerprints to a corresponding map.
It would be nice to see an overview of oncological applications (could be limited to the cancer types explored) to which MRF has been applied. I would suggest using a table. The authors could list the specific methods, sample sizes, performances and possible limitations/considerations.
The authors mention different techniques that have been combined with MRF, such as CEST. It would be nice to add an overview as to which combinations offer which benefits and limitations with regard to tumor types. What is the pathophysiology examined? What is the clinical potential? (compare to Liu et al. doi: https://doi.org/10.1016/j.jcmg.2018.08.028, Central Illustration)
Oncologist and clinical radiologist will be interested in details such as, are there only vendor MRF packages, or do vendor neutral options exist? Is reconstruction on standard vendor's computational hardware feasible?
Specific comments:
- If the authors are truly aiming at increased accessibility of the review to clinicians with minimal radiological background, I would suggest keeping the introduction simpler (compare to Poorman et al. doi: https://doi.org/10.1002/jmri.26836)
- Line 66-67: hypoxia would be defined by inadequate tissue oxygenation, I would suggest simply using “tissue oxygenation”
- Line 179-185: Perfusion-weighted magnetic resonance imaging techniques, such as dynamic contrast-enhanced MRI (DCE-MRI) and dynamic susceptibility contrast-MRI (DSC-MRI) have the potential to overcome the shortcomings of conventional MRI and help distinguish tumor from treatment‐induced processes such as pseudoprogression and pseudoresponse. They play an important role in treatment evaluation of glioma patients and are readily available at most specialized centers. Quantification of perfusion images remains the largest challenge to overcome for standardization of perfusion MRI in imaging protocols. (e.g. van Dijken et al. doi: 10.1002/jmri.26306)
Suggesting that MRF could be a “simple” and “readily available” method neglects the need of advanced post-processing methods with computational challenges including the need for sufficient storage space and the limiting factor of reconstruction time. - Line 188-189: Correct would be “Lesional T2-Mapping has been associated with the early detected of tumour progression under anti-angiogenic therapy”
- Line 201: Up to date there is no clinical evidence that directly links gadolinium retention to adverse health effects in patients with normal kidney function. If anything, you should use “potentially harmful”.
- Line 554: Is the assumption of a possible clinical implementation within 10 years based on anything? Are there explicit effort e.g. by the radiological societies’ study groups for large multi-center trials?
Round 2
Reviewer 2 Report
The authors have made a good effort to improve their manuscript and responded to all my points in my initial review. Particularly I appreciate the inclusion of Figure 6 and the rewritten introduction.